# Sequence dependence of transient Hoogsteen base pairing in DNA

**Alberto Pérez de Alba Ortíz**[1,2], **Jocelyne Vreede**[1], **Bernd Ensing**[1,3]*

**1** Van 't Hoff Institute for Molecular Sciences and Amsterdam Center for Multiscale Modeling, University of Amsterdam, Amsterdam, The Netherlands, **2** Soft Condensed Matter, Debye Institute for Nanomaterials Science, Utrecht University, Utrecht, The Netherlands, **3** AI4Science Laboratory, University of Amsterdam, Amsterdam, The Netherlands

* b.ensing@uva.nl

**Data Availability Statement:** All the data and PLUMED input files required to reproduce the results reported in this paper are available on PLUMED-NEST (www.plumed-nest.org), the public

## Abstract

Hoogsteen (HG) base pairing is characterized by a 180˚ rotation of the purine base with respect to the Watson-Crick-Franklin (WCF) motif. Recently, it has been found that both conformations coexist in a dynamical equilibrium and that several biological functions require HG pairs. This relevance has motivated experimental and computational investigations of the base-pairing transition. However, a systematic simulation of sequence variations has remained out of reach. Here, we employ advanced path-based methods to perform unprecedented free-energy calculations. Our methodology enables us to study the different mechanisms of purine rotation, either remaining inside or after flipping outside of the double helix. We study seven different sequences, which are neighbor variations of a well-studied A·T pair in $A_6$-DNA. We observe the known effect of A·T steps favoring HG stability, and find evidence of triple-hydrogen-bonded neighbors hindering the inside transition. More importantly, we identify a dominant factor: the direction of the A rotation, with the 6-ring pointing either towards the longer or shorter segment of the chain, respectively relating to a lower or higher barrier. This highlights the role of DNA's relative flexibility as a modulator of the WCF/HG dynamic equilibrium. Additionally, we provide a robust methodology for future HG proclivity studies.

## Author summary

Recently, an alternative DNA base-pairing conformation, known as Hoogsteen (HG), has been found to coexist with the well-known Watson-Crick-Franklin (WCF) pairing. Several experimental and computational studies have focused on this heterogeneity, as it is involved in various recognition and replication processes. The WCF-to-HG transition mechanisms consist of a ±180˚ rotation of the purine base, occurring either *inside* of the double helix or while flipping temporarily *outside* of it. Even though molecular dynamics simulations can provide fine details about the transition pathways and their free-energy barriers, the computational cost has limited most studies to focus on only one particular chain ($A_6$-DNA). Here, we investigate the sequence dependence of the base-pairing transition, by systematically varying the direct neighbors of a transitioning A·T pair; probing inside and outside pathways in seven distinct systems. We discover that triple-hydrogen-

repository of the PLUMED consortium, as plumID:21.033.

**Funding:** Simulations were performed on the carbon cluster at the University of Amsterdam and on the Dutch national e-infrastructure with the support of SURF Cooperative (SURFSARA) [2020.015]. A.P.A.O. received funding and a salary from the Mexican National Council for Science and Technology (CONACYT) [327780 / 382262]. Funding for open access was provided by the University of Amsterdam. The funders had no role in study design, data collection and analysis, decision to publish, or preparation of the manuscript.

**Competing interests:** The authors have declared that no competing interests exist.

bonded neighboring base-pairs hinder the inside rotation mechanism, due to the reduced flexibility needed for internal base rotation. Across all sequences, we confirm that outside transitions have a lower free-energy barrier. Most importantly, we observe that the direction of the A rotation, with the A 6-ring pointing either towards the long or short end of the modelled DNA chain, has a determinant effect on the height of the free-energy barrier. These results point to a critical role of DNA's small- and medium-scale flexibility in modulating the proclivity of HG base pairs; providing a handle that might be employed by several biological mechanisms.

## Introduction

In 1953, emblematic studies by Watson and Crick [1], and Franklin and Gosling [2] defined the structure of DNA for the first time. The discovery of a specific pairing of purine and pyrimidine bases via hydrogen bonds revealed not only DNA's double-helical shape, but also the basis for its replication, which is essential for a genetic information carrier. In Fig 1A we depict a Watson-Crick-Franklin (WCF) A·T base pair, with the characteristic hydrogen-bonding pattern. Six years later, in 1959, Hoogsteen proposed an alternative base-pairing motif based on crystallographic data from A·T crystals, in which the purine base *rolls* 180˚ around the glycosidic bond [3], i.e. from *anti* to *syn*, with respect to the WCF geometry, as depicted in Fig 1B. In the Hoogsteen (HG) configuration, the pyrimidine forms hydrogen bonds with the 5-ring of the purine, rather than with its 6-ring, causing the opposite backbone C1' atoms of the bases to be at a somewhat shorter distance, as well as some degree of twisting and bending of the double helix in the vicinity of the base pair [3, 4]. In the last decades, a number of studies have shown that the abundance of the HG conformation is non-negligible in canonical duplex DNA, and that its biological implications are very relevant. In 2011, Nikolova and coworkers reported the transient presence of HG base pairs in specific steps inside canonical duplex DNA using nuclear magnetic resonance (NMR) relaxation dispersion spectroscopy [5]. They reported populations of A·T and G·C HG base pairs of around 0.5%, with residence times of up to 1.5 ms. Recent studies have measured even larger HG populations, of 1.2%, in an A·T rich segment [6]. A few years later, Alvey and colleagues, also using NMR relaxation dispersion, demonstrated that HG base pairs appear in more diverse sequences than previously expected [7]. These studies shifted the perspective on HG base pairs, which were initially thought to appear mainly in distorted or damaged DNA, but are now considered to coexist with the WCF motif in a dynamic equilibrium [8]. A recent survey of structures of DNA-protein complexes in the Protein Data Bank (PDB) showed 140 HG base pairs out of a total of around 50,000 [4]. Some of these complexes are of particular biological relevance. For example, the human DNA polymerase-$\iota$ performs replication exclusively via HG base pairing [9, 10], a function that was previously thought to be unique of WCF base pairs. Other examples of the involvement of HG in DNA-protein complexes include the p53 tumor suppressor protein [11], the TATA-box binding protein involved in transcription [12], and the MAT$\alpha$2 homeodomain which regulates transcription in cells [13].

   The newly discovered relevance of HG base pairs, and the dynamic equilibrium in which they coexist with WCF base pairs, demands a more detailed understanding of the transition mechanisms between both conformations. Given the difficulty of observing the short-lived intermediate states experimentally, computational approaches based on molecular dynamics (MD) simulations, boosted with enhanced sampling, have provided valuable insights. In their original work from 2011, Nikolova and co-workers complemented their NMR results with

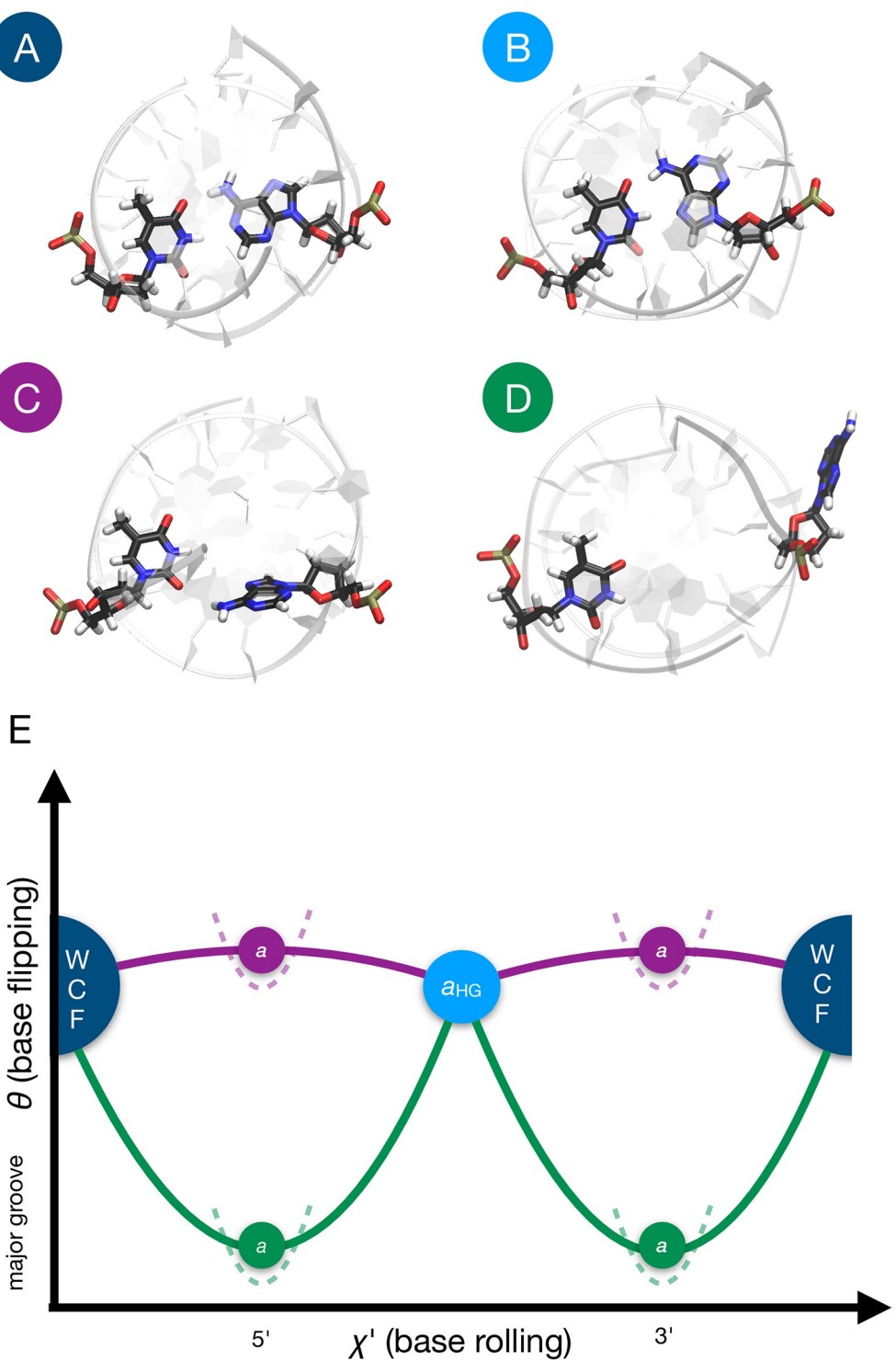

**Fig 1. WCF-to-HG conformations and pathways. (A)** WCF conformation of the A16·T9 base pair in $A_6$-DNA. **(B)** HG conformation of the same base pair. **(C)** Intermediate state of the WCF-to-HG transition via the *inside* pathway, i.e. with the rolling of A16 occurring within the double helix. **(D)** Intermediate state of the WCF-to-HG transition via the *outside* pathway. i.e. with A16 flipped out of the double helix via the major groove **(E)** Two cyclic paths in the space of the collective variables (CVs) $\chi'$ and $\theta$, which describe the rolling and the flipping of A16, respectively. Both paths start and end at the WCF state, describing a full revolution of A16 with intermediate states in which the 6-ring of A16 points toward the 5' or the 3' direction of DNA. The HG state is marked by an attractor, represented by the circle labeled $a_{HG}$. The purple path samples the inside mechanism, while the green path explores the outside mechanism.

Additional attractors, represented by the circles labeled *a*, are restrained between the WCF and the HG states along their respective paths and are also restrained along $\theta$ to prevent switching between the inside and the outside mechanisms. Restraints are represented by dashed lines.

conjugate peak refinement (CPR) simulations. They simulated the DNA sequence 5'-CGATTTTTTGGC-3' ($A_6$-DNA) in vacuo using the CHARMM27 force field [14], and studied the transition of the A16·T9 pair. Two collective variables (CVs) were used to steer the transition: (1) the $\chi$ glycosidic angle, which describes the rolling of A16 around the glycosidic bond and is defined by the atoms O4'-C1'-N9-C4; and (2) the $\theta$ base opening angle, which describes the flipping of A16 outside of the double helix towards the major groove and is defined in [15]. The CPR simulations revealed two types of pathways: one in which the rolling of the adenine, either clockwise or counterclockwise, occurs within the double helix, i.e. with a small base opening angle; and another in which the rolling occurs outside of the double helix, i.e. with a large base opening angle. We refer to these two types of pathways as *inside* and *outside*, respectively. Examples of intermediate states of the two types of pathways are shown in Fig 1C and 1D. The ($\chi$, $\theta$) pseudo-free-energy landscapes of Nikolova and coworkers showed an overwhelming preference for the inside mechanism, likely influenced by the lack of solvent to stabilize the flipped conformations. In 2015, Yang and colleagues used umbrella sampling (US) to obtain a ($\chi$, $\theta$) free-energy landscape of the same A16·T9 pair in $A_6$-DNA [16]. They employed a version of the AMBER99-BSC0 force field [17] with modified parameters for the glycosidic torsion, and used explicit TIP3P water [18]. Their vast US calculations, with over 300 windows, revealed multiple pathways, including inside ones, and outside ones opening either towards the major or the minor groove. Yang and coworkers reported a free-energy difference from the WCF to the HG state of 4.4 kcal/mol, close to Nikolova and coworkers' NMR result of 3.0 kcal/mol. The free-energy barrier was of 10–11 kcal/mol for the inside pathways, and up to 14 kcal/mol for the outside ones, again close to NMR results of 16 kcal/mol. A few years later, Yang and coworkers reported another ($\chi$, $\theta$) free-energy surface, this time for a base pair with an N1-methylated adenine [19]. Similarly, Ray and Andricioaei calculated a ($\chi$, $\theta$) free-energy surface for DNA and RNA [20]. While such 2D free-energy landscapes are rich in insight, they also demand significant computational expense to obtain several $\mu$s-long MD runs; hindering the systematic study of multiple DNA variations. In particular, investigations of the effect of the DNA sequence on HG base-pairing proclivity have remained beyond computational reach, even though there is experimental evidence that specific sequence patterns, e.g. A·T steps, can favor HG base pairing [21].

Path-based approaches offer an efficient alternative to study the WCF-to-HG transition. Techniques like path-metadynamics (PMD) [22]—i.e. metadynamics using as a unique CV the progress along an adaptive path connecting two known states in a multidimensional CV-space—focus sampling on the transition channels, rather than on the entire conformational landscape, e.g. the entire ($\chi$, $\theta$) plane. Recently, we performed a PMD mechanistic study [23], where we modeled the known $A_6$-DNA with the state-of-the-art AMBER99-BSC1 force field [24] in explicit solvent. We focused on one direction of rotation, with the A16 6-ring pointing toward the A17 neighbor in the 3' direction of DNA. We analyzed additional CVs, such as: the donor-acceptor distances of the characteristic hydrogen bonds of each type of base pairing, the distance between the backbone C1' atoms, and the distance between the neighboring bases of the rolling A16 (C15 and A17). The latter CV displayed an increase at the mid-rotated intermediate state of inside pathways, indicating that the neighboring bases are displaced in order to accommodate the rotation of the adenine. In contrast, the distance between bases neighboring the rolling A16 remained unaffected in outside transitions, indicating that the inside pathways

might be more subject to sequence dependence than the outside ones. We also found that, while the $\theta$ pseudo-dihedral as defined in [15] is effective at biasing base opening, the $\chi$ dihedral can induce rotations of the sugar ring, rather than of the adenine. This can be addressed by defining a less locally dependent pseudo-dihedral based on centers of mass, $\chi'$, as done in [25]. Most importantly, and contrary to previous studies, we found a less pronounced preference for the inside pathway [23]. Together with other coauthors, we recently reported another pathway-focused study [25]. We used transition path sampling (TPS) [26] to generate unbiased trajectories, and transition interface sampling (TIS) [27] to calculate a WCF-to-HG free-energy difference of 3.2 kcal/mol, which compares well with Nikolova and coworker's previous experiments [5]. Most importantly, TPS showed trajectories spontaneously changing from the inside to the outside mechanism, evidencing a preference for the latter one. This raises a debate with the previous results from free-energy surfaces, which favored inside pathways, although having either no explicit solvent [5] or restrained neighboring bases in their setup [16]. A predominance of outside pathways could explain why sequence variation does not seem to impact significantly the WCF-to-HG free-energy barrier, according to Alvey and coworkers' NMR results [7]. Markov state modeling has also confirmed a dominant outside pathway [20].

In this work, we apply recent advances in PMD—which allow to treat multiple pathways in parallel—to study the WCF-to-HG inside and outside mechanisms in diverse DNA sequences with high efficiency. Unlike standard metadynamics [28], PMD does not suffer from exponential performance scaling with the number of CVs [29]. This is because the sampling is effectively done in only 1D, i.e. on the normalized progress component along the path, $s$, connecting the two known stable states in CV-space. This sampling along the path can be done with other well-established methods, like US, or using common algorithmic extensions, such as multiple walkers [23, 30]. The path curve is optimized based on the restrained cumulative sampling density, which is an estimator of the free-energy gradient, such that the method converges to the closest low free-energy path from the initial guess. However, this optimization can be challenging in systems with multiple pathways, especially when the transition can switch between mechanisms, as seen for the WCF-to-HG inside and outside pathways [23, 25]. To tackle this scenario, we recently developed multiple-walker multiple-path-metadynamics (MultiPMD) [31]. In this scheme, we initialize several paths, each with an associated group of walkers. Under normal conditions, all paths would converge to the same low free-energy channel. By introducing special walkers, i.e. *repellers* or *attractors*, the paths can be forced to diverge and find alternative mechanisms connecting the known states. Thus, with MultiPMD one can calculate the free-energy profiles along multiple pathways simultaneously, with sub-exponential performance scaling with the number of CVs, and exploiting the parallelism of current supercomputing resources.

Here, we apply MultiPMD to find inside and outside pathways, and free-energy profiles, for the WCF-to-HG transition of the A16·T9 pair in seven sequence variations based on the well-studied $A_6$-DNA sequence. In order to verify the effect of local variations, we test four different nucleobases (A,T,G,C) as direct neighbors on the 5' side of the rolling A16. We repeat the procedure for the neighboring base pair on the 3' side. Our MultiPMD methods provide, for the first time, sufficient computational agility for a systematic evaluation of HG base pairing in multiple DNA chains. This enables us to extract trends about the structure and free energy of the base-pairing transition across sequences.

## Materials and methods

### Sequence variations

From [5] we take the original $A_6$-DNA sequence 5'-CGATTTTTTGGC-3', with its complementary strand 3'-GCTAAAAAACCG-5', as shown in Fig A in S1 Text. We study the WCF-

to-HG transition of the A16·T9 base pair, which is produced by the 180˚ rotation of A16 around its glycosidic bond. The original direct neighbors of A16 are C15 and A17, in the 5' and in the 3' direction respectively, which gives the local environment CAA. To test the local effect of the direct neighbor of A16 in the 3' direction, we replace A17 with T, G and C. This yields the new local environments CAT, CAG and CAC. We apply the same criteria for the direct neighbor of A16 in the 5' direction, and replace C15 with A, T and G. This gives the new local environments AAA, TAA and GAA. All sequences are generated as ideal B-DNA duplex structures using the make-na tools [32], and can be regenerated with the current W3DNA server [33]. Since make-na generates WCF base pairs, we manually rotate A16 to obtain the HG state for each sequence.

## Molecular dynamics

System preparation is done with GROMACS 5.4.1 [34]. We employ the AMBER99-BSC1 force field [24] to model the DNA's interatomic interactions. Each sequence is placed in a periodic dodecaedron box, with a distance of 1 nm between the DNA sequence and the edge of the box. Each system is then solvated in water, which is modeled by the TIP3P [18] force field. We add 25 mM of NaCl—modeled with the AMBER99 force field [35]—to mimic physiological conditions and neutralize the charge of each system. The systems are energy minimized at each stage of the preparation process. To run MD, the canonical sampling through velocity rescaling (CSVR) thermostat [36] is set at at 300 K, and the Parrinello-Rahman barostat [37] at 1 bar. Each sequence, both at the WCF and at the HG state, is equilibrated for 100 ns with a time step of 2 fs. From the equilibrations, we analyze key structural features and their variations from sequence to sequence.

## Multiple-path-metadynamics

MultiPMD production runs are performed with GROMACS 5.4.1 [34], patched with PLUMED 2.3.1 [38] and with the added PMD code, available at: https://github.com/Ensing-Laboratory/PathCV. We use two CVs: the base opening pseudo-dihedral angle $\theta$, as defined in [15], which has been repeatedly proven effective to bias base flipping [5, 16, 23]; and the base rolling pseudo-dihedral angle $\chi'$, as defined in [23], which is a correction to the glycosidic torsion $\chi$ to prevent sugar rotation. The fitness of these two CVs has been discussed in [39]. A value of $\chi' = 0$ rad implies a mid-rotated A16 with its 6-ring pointing in the 3' direction, while $\chi' = \pm\pi$ implies that the A16 6-ring points toward the 5' direction. Negative values of $\theta$ imply base flipping toward the major groove. We do not consider base flipping toward the minor groove, since our previous TPS study showed spontaneous switching only from the inside to the major groove pathway [25]. To handle both directions of rotation of A16, the $\chi'$ angle is treated with its sine and cosine. Then, the paths are curves in the CV-space spanned by [cos($\chi'$), sin($\chi'$), $\theta$]. These paths are cyclic, starting and ending at the WCF configuration, whose coordinates in the [cos($\chi'$), sin($\chi'$), $\theta$]-space are determined by averaging the CVs from the corresponding equilibration run. The initial guess for all paths describes a full revolution of A16, transitioning from WCF to HG to WCF, with no flipping. All outside paths are discretized as strings with 39 nodes, with the initial and the final node fixed at the same point in the [cos($\chi'$), sin($\chi'$), $\theta$]-space, which marks the WCF state. The inside paths are discretized as strings of 11 nodes, since they require less flexibility. The progress component along the path, $s$, which usually grows from $s = 0$ to $s = 1$ from the initial to the final node, is now set to grow from $s = -1$ to $s = +1$, and to be periodic in the same range, in order to match the cyclic nature of the path. This implies that the WCF state corresponds to $s = \pm 1$. on the other hand, the HG state corresponds to $s \approx 0$, but the exact value cannot be known a priori. Assigning the HG

state to a specific fixed node in the middle of the path would require to make an assumption about the length of each section of the path. Instead, the HG state is marked by an attractor, i.e. a walker that does not participate in the free-energy calculation. The HG attractor is harmonically restrained, with a force constant of 50 kcal/mol, at the average value of $\chi'$ and $\theta$ during the corresponding equilibration run. Values of $-1 < s < 0$ imply a rotation with the A16 6-ring in the 5' direction, while values of $0 < s < +1$ signify a rotation in the 3' direction.

For each sequence, we initialize two paths, one for the inside, and one for the outside mechanism. We only consider the outside mechanism with opening toward the major groove, as previous work already demonstrated that opening towards the minor groove is unlikely [16, 25]. As we reported in [23], switching between the inside and the outside mechanisms can occur during a PMD calculation of base rolling. Since path-switching prevents convergence, we use a new strategy to keep the paths apart. The separation is induced by special walkers, called attractors, that do not take part in the free-energy calculation and guide the paths through specific intermediate states. We add two attractors on each path, which are steered to $s = 0.5$ and $s = -0.5$ by a moving harmonic restraint with a force constant of 5000 kcal/mol per squared path unit during the first 20 ps of the simulation, such that they are located at intermediate states in the WCF-to-HG and the HG-to-WCF sections of the cyclic path. These intermediate states between the two kinds of base pairing have the A16 mid-rotated, perpendicular to the other bases. To keep the inside path from flipping, its two attractors are restrained by a harmonic potential with a force constant of 5000 kcal/(mol rad$^2$) at $\theta = 0.0$, which prevents them from leaving the confines of the double helix. In turn, the repellers of the outside path are restrained by a harmonic potential with a force constant of 5000 kcal/(mol rad$^2$) at $\theta = -\pi/2$, which ensures that the A16 stays flipped toward the major groove at the mid-rotation. The large values for all the force constants are chosen because the attractors are located near the top of the free-energy barriers, and we require them to remain in these high-energy, mid-rotated, conformations. Then, each path has one attractor at the HG state, and two attractors at intermediate states; one at $s = 0.5$ and one at $s = -0.5$. In Fig 1E, we show a scheme of the inside and outside paths, the fixed nodes, the HG attractor and the intermediate-state attractors.

Nine standard walkers—for a total of 12 walkers per path—perform metadynamics on the $s$ component of the path. The metadynamics Gaussian potentials have a width of 0.1 path progress units and a height of 0.05 kcal/mol, and are deposited every 1 ps. The paths are updated every 1 ps. The value of the half-life parameter is infinite for inside paths and 20 ps for outside paths. This parameter determines the flexibility of the path by setting the amount of simulation time that it takes for previous samples to weight only 50% of their original value for path updates. We use a *tube* potential—i.e. an upper harmonic wall at a distance of 0.0 on the component perpendicular to the path, $z$—with a force constant of 50 kcal/mol per path unit to maintain all walkers near the path. For all the sequences, we analyze the adapted paths after 7 ns of sampling. We verify that the dynamics have reached the "free-diffusion" regime along the biasing coordinate, indicating that the free-energy profile has been reasonably approximated. For all sequences, this occurs before 2 ns of biasing. We obtain metadynamics-based free-energy profiles—i.e., the negative of the sum of Gaussian potentials—every 0.1 ns, starting at 2 ns and finishing at 7 ns. We obtain our final free-energy profiles and error bars from the average and standard deviation of these metadynamics-based estimations. There are a few exceptions to the general protocol, which we detail in S1 Text.

All the data and PLUMED input files required to reproduce the results reported in this paper are available on PLUMED-NEST (www.plumed-nest.org), the public repository of the PLUMED consortium [40], as plumID:21.033.

## Results and discussion

### Stable-state structures

From the 100 ns equilibration runs at the WCF and HG stable states, we analyze structural variations between the different sequences. Table A in S1 Text presents the definitions, averages and standard deviations of several CVs during the equilibrations. See [23] and Fig C in S1 Text for more details about the CVs. First, we note that all equilibrations are stable at the respective base-pairing configurations, as evidenced by the characteristic hydrogen-bond distance, $d_{\mathrm{WCF}}$ or $d_{\mathrm{HG}}$, and the conserved $d_{\mathrm{HB}}$ with average values of $\sim 3$ Å; and by the base rolling angle $\chi'$ close to either +1.5 rad in WCF, or to -1.5 rad in HG. However, the standard deviations of $d_{\mathrm{HG}}$ for the HG states, which range from 0.2 to 0.7 Å, already indicate that the stability of HG base pairing is not equal for all sequences. In contrast, the standard deviation of $d_{\mathrm{WCF}}$ for WCF base pairs is of $\sim 0.1$ Å, indicating more rigid hydrogen bonds. Another structural signature of the two kinds of base pairing is the distance between the C1' atoms of A16 and T9, $d_{\mathrm{CC}}$, which shows an expected constriction from $\sim 10.6$ in WCF, to $\sim 9.1$ Å in HG conformations [4]. A key feature that varies significantly from sequence to sequence is the distance between the neighboring bases, $d_{\mathrm{NB}}$, with a range from 7.4 to 8.0 Å. In the following sections, we show how this CV relates to the free-energy barriers from the WCF-to-HG transition.

### Free-energy differences and barriers

Inside and outside pathways, together with their corresponding free-energy profiles, are shown in Fig B in S1 Text. For all free-energy profiles, we report error bars. Additionally, we change the sampling time by 2 ns and show that the free energies do not change beyond the error bars (See dotted lines in Fig B in S1 Text). The attractors successfully keep the paths separated, with the inside ones near $\theta = 0$ and the outside ones flipping toward the major groove. The larger error bars for the outside paths of sequences GAA and CAT indicate that, unlike the other sequences, their optimal outside transition channel might not intersect with the attractor and have a different base opening angle. Note that our attractors mark only two landmarks in a continuous spectrum of inside-to-outside pathways with different barriers, which overlap at the stable states. We do not relocate the attractors in order to maintain a direct comparison with the rest of the sequences. The free-energy profile for the CAA sequence compares well to our results from [23] and [39], which consider only the 3' direction of rotation. To simplify the analysis of the numerous free-energy profiles, in Fig 2 we show only the barriers, in both the 3' and the 5' directions of rotation, as well the free-energy difference from WCF to HG. The free-energy differences calculated for the same sequence along inside and outside paths are consistent within $\sim 2$ kcal/mol and have overlapping error bars; providing a sensible check for our profiles (see Fig 2B). The small inconsistencies are due to the implicit error of the metadynamics estimation and to the tube potentials, which restrain the sampling of the free-energy valleys in the direction perpendicular to the paths, thereby modifying the distributions with respect to unbiased sampling of the minima. As expected, WCF base pairing is preferred over HG in all cases. The free-energy difference between both states ranges from $\sim 0.5$ to $\sim 6$ kcal/mol across all sequences. This range agrees with free-energy differences obtained by NMR relaxation dispersion for other sequences [7]. According to our calculations, the sequence TAA presents the most favored HG state, with a free-energy difference of 0.6 to 2 kcal/mol with respect to WCF; and with the lowest barrier overall (9.7 kcal/mol) via the outside 3' rotation. This result agrees with experimental reports of HG stability being increased by A·T steps [21]. The CAT sequence shows the most disfavored HG state, with a difference of 4.9 to 5.9 kcal/mol and the highest barrier overall (20 kcal/mol) via the 5' inside rotation.

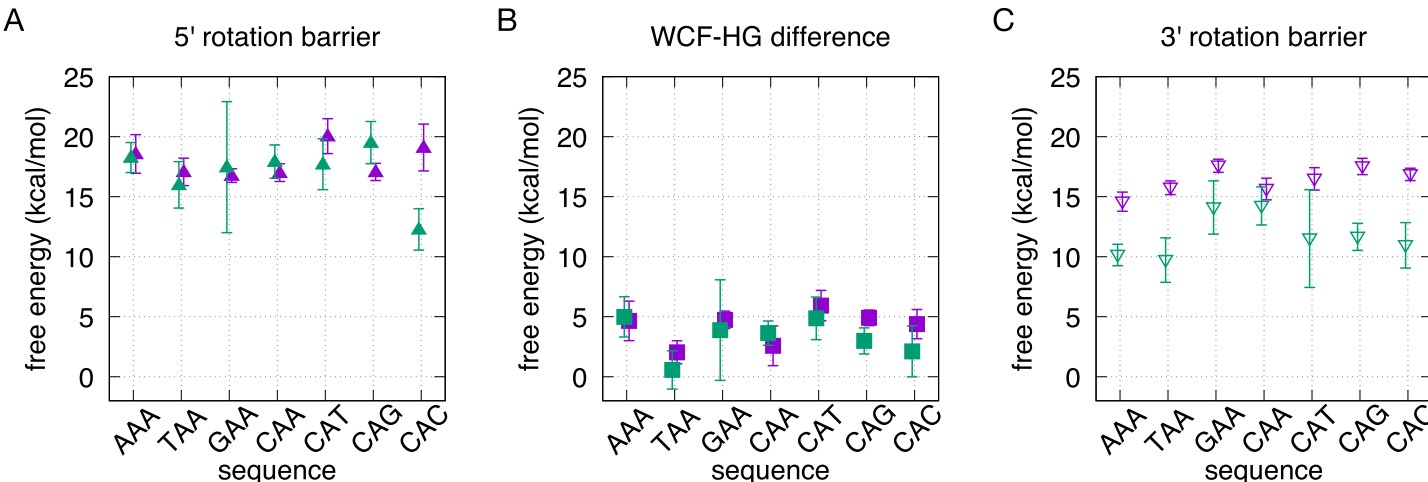

**Fig 2. WCF-to-HG free-energy differences and barriers for the seven sequences.** The sequences are shown in Fig A in S1 Text. The free-energy differences and barriers are extracted from the average free-energy profiles and error bars in Fig B in S1 Text. Inside paths are shown in purple and outside paths are shown in green. **(A)** Free-energy barriers via the rotation with the 6-ring of A16 pointing in the 5' direction. **(B)** Free-energy differences. **(C)** Free-energy barriers via the rotation with the 6-ring of A16 pointing in the 3' direction.

For all sequences, the lowest free-energy barrier is that of the outside pathway with a 3' rotation (see Fig 2C). Such a prevalent outside mechanism agrees with our previous results for the CAA sequence using TPS [25], as well as Markov state modeling by Ray and Andricioaei [20]. Most of the outside 3' rotations also show a transition state closer to HG ($s \approx 0$) than to WCF, which agrees with experiments [7]. These results imply that the conformational penalty of A16 rolling within the double helix is higher than that of it flipping out and back in.

## Sequence dependence of the free-energy barriers

In [23], we showed how the inside rotation induces an increase in the distance between the neighbors of A16, which suggests that inside pathways have a stronger neighbor-dependence. Specifically, one may hypothesize that less flexible—i.e. double-ringed (A,G) or triple-hydrogen-bonded (G,C)—neighboring bases can hinder the inside rotation of A16. We observe such a trend in the barriers for the 3' inside rotation (see Fig 2C). The 5' neighbor variations with respect to CAA yield the following ranking of sequences, from the highest to the lowest barrier: GAA, CAA, TAA, AAA. While the 3' neighbor variations yield the following order, again for the highest to the lowest barrier: CAG, CAC, CAT, CAA. These rankings would indicate that the barriers are increased mostly by G neighbors, followed by C, T and A. However, this trend is not repeated for the inside rotations in the 5' direction (see Fig 2A), or for the outside rotations. Instead, we notice that the most dominant influence on the free-energy barrier is the direction in which the 6-ring of A16 points during the rotation. Remarkably, the rotation in the 3' direction has a consistently and significantly lower barrier than in the 5' direction (see Fig 2A and 2C). This is due to the asymmetrical length of the DNA sequence in each direction (see Fig A in S1 Text), i.e. the position of the rolling base along the sequence. Starting from the rotating base, A16, there are three base pairs in the 5' direction and eight in the 3' direction, which imply a difference in flexibility. We observe that, when the 6-ring rotates in the 5' direction, the 5-ring protrudes in the opposite direction toward the longer segment of DNA, which is less flexible, causing a higher barrier. In contrast, when the 6-ring rotates in the 3' direction, the 5-ring pushes toward the shorter, more flexible, segment of DNA, causing a lower barrier.

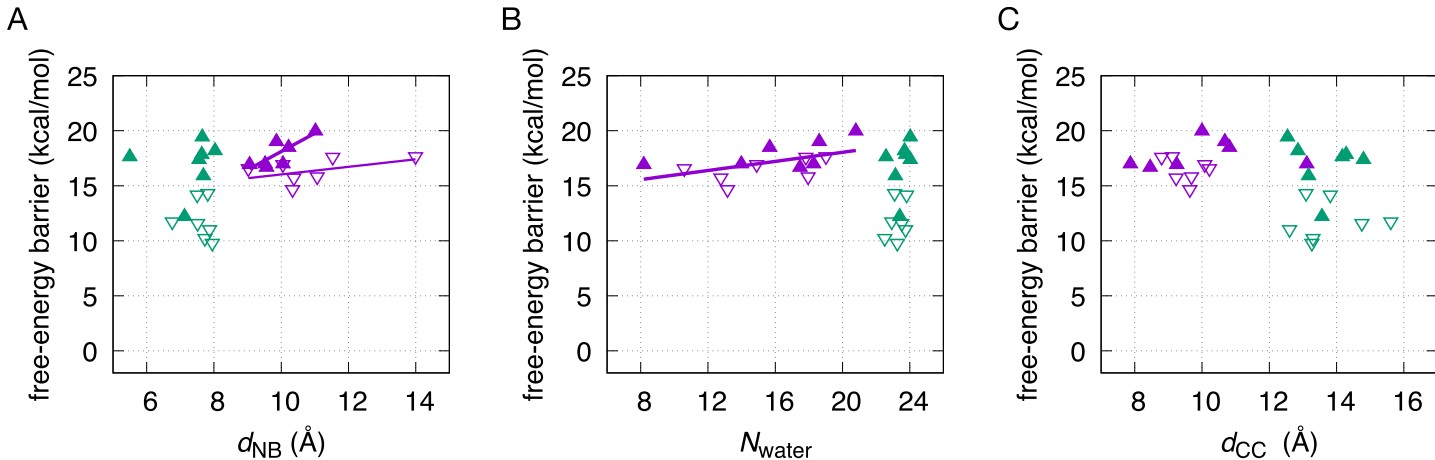

**Fig 3. Trends between the WCF-to-HG free-energy barriers and the average value of a few selected CVs.** The free-energy barriers are reported in Fig 2. The average values of the selected CVs are taken from the attractors restrained at $s = -0.5$ and $s = 0.5$. Inside paths are shown in purple and outside paths are shown in green. The barriers for the rotation with the A16 6-ring pointing in the 5' direction are represented in solid triangles pointing up, while the ones in the 3' direction are represented in outlined triangles pointing down. **(A)** Free-energy barrier vs. the distance between the neighboring bases of A16, $d_{NB}$. The fit of the inside 5' direction barriers is shown with a thick purple line (free-energy barrier = 1.7 kcal/(mol Å)$d_{NB}$ + 1.5 kcal/mol). The linear fit of the inside 3' direction barriers is shown with a narrow purple line (free-energy barrier = 0.3 kcal/(mol Å)$d_{NB}$ + 12.6 kcal/mol). **(B)** Free-energy barrier vs. the number of water oxygens within 6 Å of the N6 atom of A16, $N_{water}$. The linear fit of the inside 5' and 3' direction barriers is shown with a thick purple line (free-energy barrier = 0.2 kcal/mol $d_{NB}$ + 13.9 kcal/mol). **(C)** Free-energy barrier vs. the distance between the C1' atoms of A16 and T9, $d_{CC}$.

We analyze trends between the free-energy barriers and a few significant CVs. The average values of the CVs are taken from the attractors restrained at the intermediate states $s = -0.5$ and $s = 0.5$, which are close to the peaks of the free-energy barriers. In Fig 3A, we show the distance between the two neighbors of the rolling A16, $d_{NB}$, in relation to the free-energy barrier. Several trends can be identified. First, $d_{NB}$ remains at low values ($< 8$ Å) for all outside paths, close to those of the stable states (see Table A in S1 Text); confirming that the transition with base flipping induces much less deformation of the neighbors. On the other hand, all the inside paths show larger values of $d_{NB}$, ranging from $\sim 9$ to $\sim 14$ Å. Among the inside paths, two distinctive trends can be observed. Inside rotations in the 3' direction reach much larger values of $d_{NB}$, with a relatively low increase in the free-energy barrier (0.3 kcal/mol per Å). In contrast, for the inside rotations in the 5' direction, the free-energy barrier quickly rises (1.7 kcal/mol per Å). This again highlights the role of the DNA segment's relative length in each direction of the rolling base, as well as the favored 3' direction of rotation.

We also analyze the influence of the water solvation on the free-energy barriers (see Fig 3B). We measure the solvation using the parameter $N_{water}$ [25], i.e. the number of water oxygens within 6 Å of the N6 atom of A16, which is the atom involved in the conserved hydrogen bond of both WCF and HG base pairs. The intermediate states of the inside pathways show a wide range of $N_{water}$ values, from $\sim 8$ to $\sim 21$. For both the 3' and the 5' direction, an increase of $N_{water}$ in the intermediate state relates with an increase in the free-energy barrier (0.2 kcal/mol per water molecule). Rather than a direct solvation of A16, the increase of $N_{water}$ in the inside pathways is due to the separation of the neighboring nucleotides, as measured before by $d_{NB}$, which generates an opening accessible to water. On the other hand, outside pathway intermediates present larger and mostly constant values of $N_{water} \approx 24$, with the flipped conformations of A16 being stabilized by the water solvent. Our reported ranges of $N_{water}$ for inside and outside mechanisms agree with those in [25].

Additionally, we study the relation of the free-energy barrier with the distance between the C1' atoms of A16 and T9, $d_{CC}$ (see Fig 3C). Inside pathway intermediates show values of $d_{CC}$

mostly from $\sim 8$ to $\sim 11$ Å. This spans a larger range than that of the stable states ($\sim 9.1$ to $\sim 10.6$ Å), indicating another possible conformational penalty for the inside pathways, but there is no clear correlation with the free-energy barrier. The intermediates of the outside pathway show larger $d_{CC}$ values, from $\sim 12$ to $\sim 16$ Å, which are expected given that the A16·T9 base pair is completely broken. There is no clear correlation of the free-energy barriers via the outside pathways with $d_{CC}$, or with the other analyzed parameters.

## Conclusion

Our efficient MultiPMD protocol allows, for the first time, a systematic study of HG base-pairing proclivity in diverse DNA chains. We investigate the rotation of the A16-T9 base pair in seven sequences, based on the previously studied $A_6$-DNA [5, 23, 25]. The seven sequences are variations of the direct neighbor of the rolling A16 in the 3' and the 5' direction of DNA (see Fig A in S1 Text). For all sequences, we study the inside and outside pathways, i.e. without and with base flipping, as depicted in Fig 1. We only consider the outside pathway with flipping toward the major groove, because our previous TPS study showed spontaneous opening only in that direction [25]. However, opening toward the minor groove could be included in a future study by adding a third path and set of attractors. We obtain WCF-to-HG free-energy profiles for the inside and the outside pathway of each of the seven sequences (see Fig B in S1 Text). Our profiles are validated by: 1) the previous result for the CAA sequence reported in [39]; 2) the consistent WCF-to-HG free-energy difference between our inside and outside calculations; and 3) the relatively favored HG base pairing for the TAA sequence, which agrees with experimental evidence about the effect of A·T steps [21]. We run $\sim 7$ ns with twelve walkers to calculate each free-energy profile; placing our total runtime to obtain the inside and outside profiles of one sequence at $\sim 168$ ns. This requirement is well below the $\mu s$-long runs required for calculating previous free-energy surfaces [16, 19]. Our 1D free-energy profiles along the path progress, $s$, could be compared with 2D free-energy surfaces by a posteriori optimizing a string on the ($\chi'$, $\theta$) plane, as described in [41]. To this end, we have made a post-processing tool for standard metadynamics free-energy surfaces available at: https://www.compchem.nl/software_package/trace-irc/.

From our free-energy calculations, we observe that all sequences have a preferred outside pathway, in which the A16 rotates with its 6-ring pointing in the 3' direction (see Fig 2). This dominant outside pathway agrees with our previous results using TPS [25], as well as with published reports using Markov state models [20]. Our result is likely to settle the debate arising for previous simulations that showed favored inside pathways, but either with no explicit solvent [5], or with a modified force field and restrained neighboring bases upon initialization [16].

We analyze the possible influence of the varied direct neighbors of A16 on the free-energy barrier for its rotation. Based on the mechanistic analysis done in [23], we expect inside paths to be more sensitive to neighbor-dependence, since they require to increase the distance between the neighbors of A16, $d_{NB}$, in order to accommodate the rotation. In Fig 2C, we observe that barriers for an inside 3' rotation are increased mostly by G neighbors, followed by C, T and A. While this could point to triple-hydrogen-bonded neighbors hindering the transition, the trend is not reproduced as prominently for outside pathways. Instead, we observe that the most impactful factor for the free-energy barrier is the direction, either 3' or 5', in which the 6-ring of A16 points during the rotation. The barriers for the rotation in the 3' direction are significantly lower than their counterparts. This difference is due to the asymmetric length of the DNA chain in each direction of the transitioning base pair. The rotation of the 6-ring in the 5' direction causes the 5-ring to protrude in the 3' direction, toward the longer

and more rigid side of the DNA chain, causing a higher free-energy barrier. In contrast, the 3'
direction of rotation causes the 5-ring to push against the shorter and more flexible side of the
DNA chain, which comes with a lower free-energy barrier. This trend is confirmed in Fig 3A.
We observe that increasing the distance between the neighbors of A16, $d_{NB}$, quickly raises the
free-energy barrier for inside rotations in the 5' direction, while the inside rotation in the 3'
direction can reach larger neighbor separations with a much lower free-energy penalty. We
also observe that $d_{NB}$ is almost undisturbed with respect to stable-state values during the out-
side transitions. In Fig 3B, we analyze the number of water molecules surrounding A16, $N_{water}$,
which is large and mostly constant for all outside intermediates. The value of $N_{water}$ for inside
intermediates is expectedly lower, and also evidences the energetically costly opening of the
neighbors already described by $d_{NB}$. Additionally, we analyze the distance between the C1'
atoms of the A16·T9 base pair, which separate significantly during the outside transitions.
Nonetheless, we do not find a CV that correlates with the free-energy barrier of the outside
pathways. In summary, our WCF-to-HG free-energy differences and barriers reveal that HG
proclivity is weakly dependent on the direct neighboring nucleotides and strongly controlled
by local flexibility. This provides a valuable filter for studies searching for HG base pairs—
which might be mislabeled as WCF—in experimentally determined structures [4, 42]. Finding
these base pairs, and their transitions pathways, is key to understand how, while WCF base
pairs store static information, HG base pairs encode dynamic functions in living organisms
[43].

We believe that this work provides a robust and efficient methodology for future investiga-
tions of HG base pairing in various sequences. This includes studies with equal number of
base pairs on both sides of the transitioning base, in order to elucidate the role of the neighbors
exclusively, without the effect of an asymmetric length. One could also study transitions of
more than one base pair, similarly to Chakraborty and Wales discrete path sampling work
[44]. More importantly, our observation about the dominant influence of the relative chain
length, and associated flexibility, towards the 5' and 3' directions of the rolling base, has major
mechanistic implications. Recent studies have observed HG base pairing in stressed, protein-
bounded DNA [42]. Protein-DNA complexes that function via HG base pairs might not only
recognize, but even induce the transition by modulating the rigidity of a DNA sequence. Our
simulation protocol can also enable a fast investigation of protein-DNA complexes. Finally,
the efficiency of the MultiPMD method also enables the use of higher levels of theory, such as
hybrid quantum mechanics/molecular mechanics (QM/MM) studies, in which the HG transi-
tion of G·C base pairs could be simulated with flexible protonation states [45].

## Supporting information

**S1 Text. The supplementary text contains further details about the simulation protocol,
collective variables and all supplementary figures and tables.**
(PDF)

## Author Contributions

**Conceptualization:** Alberto Pérez de Alba Ortíz, Jocelyne Vreede, Bernd Ensing.

**Data curation:** Alberto Pérez de Alba Ortíz.

**Formal analysis:** Alberto Pérez de Alba Ortíz.

**Funding acquisition:** Alberto Pérez de Alba Ortíz, Jocelyne Vreede, Bernd Ensing.

**Investigation:** Alberto Pérez de Alba Ortíz.

**Methodology:** Alberto Pérez de Alba Ortíz, Jocelyne Vreede, Bernd Ensing.

**Project administration:** Jocelyne Vreede, Bernd Ensing.

**Software:** Alberto Pérez de Alba Ortíz.

**Supervision:** Jocelyne Vreede, Bernd Ensing.

**Validation:** Alberto Pérez de Alba Ortíz.

**Visualization:** Alberto Pérez de Alba Ortíz.

**Writing – original draft:** Alberto Pérez de Alba Ortíz.

**Writing – review & editing:** Alberto Pérez de Alba Ortíz, Jocelyne Vreede, Bernd Ensing.

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
