## [Decision Letter · Decision Letter 0]

20 Dec 2021

Dear Dr. Ensing,

Thank you very much for submitting your manuscript "Sequence dependence of transient Hoogsteen base pairing in DNA" for consideration at PLOS Computational Biology.

As with all papers reviewed by the journal, your manuscript was reviewed by members of the editorial board and by several independent reviewers. In light of the reviews (below this email), we would like to invite the resubmission of a significantly-revised version that takes into account the reviewers' comments.

Reviewers raised serious concerns regarding the data reliability of the simulations. In the revised manuscript, the issue must be addressed in a satisfactory way. Reviewer 3 wonders about the implications for experimental biology. We realize that it is challenging to address this point in a study that addresses a fundamental question. However, we ask you to show biological relevance to concrete cases as much as possible.

We cannot make any decision about publication until we have seen the revised manuscript and your response to the reviewers' comments. Your revised manuscript is also likely to be sent to reviewers for further evaluation.

Sincerely,

Shi-Jie Chen

Associate Editor

PLOS Computational Biology

Nir Ben-Tal

Deputy Editor

PLOS Computational Biology

Reviewer's Responses to Questions

**Comments to the Authors:**

Reviewer #1: Ensing and co-workers report on the pathways and free energies of Watson-Crick to Hoogsteen base pairing in A6 DNA with multiple sequence contexts in MultiPMD (multiple-path-metadynamics) simulations exploring two collective variables that enable the conversion. Overall the manuscript is well written and well summarizes the findings from the simulations. Of some concern is the short sampling time (< 10 ns per walker), results from single (sets of) simulations , and potential bias of attracters, repellers, and force constants on the results. For example, if simulations are run twice as long per walker or with twice as many walkers are consistent results obtained? In the conclusion, it is stated you need much less simulation time that the microsecond timescales seen previously – to better justify, perhaps present results that show convergence – for example, do independent runs on same sequence (with different arrangement of ions or initial velocities) give equivalent results? Figure S2 suggests poor convergence (or high variance) in the GAA and CAT sequences.

Minor points

- What ion parameters?

- (see Fig. S1, i.e. – needs closing “)”

Reviewer #2: Perez et al. reported the in- and out-HG pairing pathways using the adaptive path sampling method. Also, they investigated sequence dependent effects on those pathways.

In this computational study, they concluded that the out-path is more favorable than in-path. This is interesting. Overall, this manuscript was well written and the results are well presented.

I recommend this manuscript be published in the PLOS comp. Bio. after some minor revisions:

1. a.The authors need to describe in detail how the free energy profile along the paths were obtained.

b.I am wondering if these free energy profiles on their optimized paths be exactly reproduced by more

conventional US-WHAM or other non-equilibrium schemes along those paths? If they are, please give a proper

reference on this.

2. In Figure 2B, the free energy difference between WC and HG in various sequences were shown. In principle, this

free difference should be the same for the inside and outside paths. But non-negligible differences (up to several

kcal/mol) were found in some sequences (CAC, CAG, ….). This may raise a concern on the reliability of this data.

Also, in GAA, the free energy difference from the outside path (in green) has too much error. The authors should

provide some reasonable explanations for these two issues.

3. For the outside paths, only the major groove opening path (negative bf angle) was considered. (In this paper, the

bf angle (theta) is defined as negative for the major groove opening, while it was defined as positive in the original

paper). Although they claim that the minor groove path is unlikely, previous papers on full 2D free energy maps

(chi, theta) showed that the minor-groove path is quite possible. By matching the two

CVs (chi', theta) used in this study, re-projection of the previously reported (chi, theta) to (chi', theta) map would

not change the popularity of the minor groove path. The authors are invited to address this difference in

discussion.

4. In Figure S1, in the 3’ variation panel, there is a typo: CAG CAG �CAT CAG?.

Reviewer #3: In this manuscript, the authors describe an enhanced sampling method, as applied to a base-flipping motion associated with the transition between canonical and Hoogsteen orientations. The primary conclusion is very intuitive: there is a smaller energetic barrier when the base first flips "out", rather than rearranging within the stacked bases of dsDNA. One certainly expects this result, since the sterics associated with base rotation in a confined environment will be extremely strong.

The manuscript is easy to read, and the work appears to be sound. However, the study is not suitable for PLOS Comp Bio. First, there is no clear connection to biology. That is, do the different barrier heights for different sequences explain a known biological phenomenon, or suggest new biological physics? As written, the results are primarily a display of the methodology. The second issue is that the general topic of enhanced sampling methods is not particular new. There are many methods available in the literature, and there is no comparison of the results with other methods. If this is a methods paper, then rigorous comparison should be provided. Comparisons would have to describe differences in accuracy and performance. This work would be more suitable for a simulation-methods-focused journal, such at JCTC.

**Have the authors made all data and (if applicable) computational code underlying the findings in their manuscript fully available?**

Reviewer #1: Yes

Reviewer #2: Yes

Reviewer #3: Yes

PLOS authors have the option to publish the peer review history of their article (what does this mean?). If published, this will include your full peer review and any attached files.

Reviewer #1: No

Reviewer #2: No

Reviewer #3: No
---

## [Decision Letter · Decision Letter 1]

19 Apr 2022

Dear Dr. Ensing,

We are pleased to inform you that your manuscript 'Sequence dependence of transient Hoogsteen base pairing in DNA' has been provisionally accepted for publication in PLOS Computational Biology.

Best regards,

Shi-Jie Chen

Associate Editor

PLOS Computational Biology

Nir Ben-Tal

Deputy Editor

PLOS Computational Biology

Reviewer's Responses to Questions

**Comments to the Authors:**

Reviewer #1: The authors largely addressed the concerns raised in previous review.

Reviewer #2: The authors have made minor revisions accordingly. This revised version appears to be sound and clear. I find this version of the manuscript to be suitable for the publication in the Plos Comp. Biol.

Reviewer #3: With the revised introduction and discussion, the context for the study is now clear. While the technical side is interesting, the revised focus on biological implications makes the study appropriate for PLOS CB.

**Have the authors made all data and (if applicable) computational code underlying the findings in their manuscript fully available?**

Reviewer #1: Yes

Reviewer #2: Yes

Reviewer #3: None

PLOS authors have the option to publish the peer review history of their article (what does this mean?). If published, this will include your full peer review and any attached files.

Reviewer #1: No

Reviewer #2: No

Reviewer #3: No

---

## [Editor Report · Acceptance letter]

20 May 2022

PCOMPBIOL-D-21-02065R1 

Sequence dependence of transient Hoogsteen base pairing in DNA

Dear Dr Ensing,

I am pleased to inform you that your manuscript has been formally accepted for publication in PLOS Computational Biology. Your manuscript is now with our production department and you will be notified of the publication date in due course.

With kind regards,

Anita Estes
